# Vimentin Plays a Crucial Role in Fibroblast Ageing by Regulating Biophysical Properties and Cell Migration

**DOI:** 10.3390/cells8101164

**Published:** 2019-09-27

**Authors:** Kristina Sliogeryte, Núria Gavara

**Affiliations:** School of Engineering and Materials Science, Queen Mary University of London, Mile End Road, London E1 4NS, UK; k.sliogeryte@qmul.ac.uk

**Keywords:** fibroblasts, cell ageing, vimentin, actin, tubulin, cell migration, cell mechanics, withaferin A, acrylamide

## Abstract

Ageing is the result of changes in biochemical and biophysical processes at the cellular level that lead to progressive organ decline. Here we focus on the biophysical changes that impair cellular function of human dermal fibroblasts using donors of increasing age. We find that cell motility is impaired in cells from older donors, which is associated with increased Young’s modulus, viscosity, and adhesion. Cellular morphology also displays parallel increases in spread area and cytoskeletal assembly, with a threefold increase in vimentin filaments alongside a decrease in its remodelling rate. Treatments with withaferin A or acrylamide show that cell motility can be modulated by regulating vimentin assembly. Crucially, decreasing vimentin amount in cells from older individuals to levels displayed by the neonatal donor rescues their motility. Our results suggest that increased vimentin assembly may underlay the aberrant biophysical properties progressively observed at the cellular level in the course of human ageing and propose vimentin as a potential therapeutic target for ageing-related diseases.

## 1. Introduction

Ageing is a complex process characterised by temporal changes in biological, biophysical and biochemical function that lead to a progressive whole-body decline throughout the lifespan of an individual. While age-related deterioration is most conspicuous at the organ level, it has been hypothesised that the underlying causes are likely to be dysfunctions at the cellular and tissue level [1]. Age is a risk factor for many pathologies, such as cardiovascular disease [2], osteoarthritis [3], idiopathic pulmonary fibrosis [4], glaucoma [5] and cancer [6]. Possibly due to the links between pathology and ageing, the majority of ageing research has focused on assessing decline in organ or tissue function and associating it to changes in genetic, epigenetic, or metabolic states. On the other hand, cellular behaviour integrates as a simpler output the plethora of molecular networks and gene up/down regulations that define the molecular state of a cell. Accordingly, complex age-associated perturbations at the molecular level may be more easily captured as aberrations at the cellular level. In spite of that, a limited number of studies have assessed age-associated changes in cell behaviour.

The majority of cellular studies on ageing have focused on recursive passaging in vitro as a surrogate of ageing in vivo [7,8,9,10]. Conversely, comprehensive studies of single cells isolated from donors at different ages have been limited and tend to focus on measuring replicative decline or the emergence of senescence within a cell population [11]. Recent studies have demonstrated that donor age can be determined using biophysical biomarkers, such as cell migration, contractility, mechanical properties and gross morphological features [12,13]. It thus follows from those findings that biophysical properties do change significantly and become aberrant as a result of donor ageing [14,15], a phenomenon that likely impairs cell function.

It is often assumed that actin is the main cytoskeletal network involved in the regulation of cell motility [16,17], the generation of contractile forces and in overall cell biophysical properties [18,19]. Recent studies have revealed that microtubules and especially the intermediate filament vimentin also play a crucial role in functions ranging from cell motility to signal transduction. Of late, vimentin has been reported to be involved in cell migration by regulating actomyosin contraction forces, interactions with the extracellular matrix, and also in the ability of the cell to move its nucleus forward [20,21]. Other studies have highlighted vimentin’s role in wound healing by coordinating fibroblast proliferation [22] or in collective cell migration by controlling traction forces [23]. Interestingly, it has been suggested that vimentin fibres are the major contributor to cytoplasmic but not cortical stiffness of cells, given that the cytoplasm of wild-type fibroblasts is twofold stiffer than that of their vimentin-deficient counterparts, even though cortical stiffness remains the same [24]. Vimentin-deficient cells also show defects in cell motility and directionality as well as a reduction in wound healing capacity [25,26] while overexpression of vimentin promotes prostate cancer cell invasion and metastasis [27]. Finally, vimentin has also been linked to ageing, with observations that senescent cells show increased levels of vimentin expression [28] and that glycation of vimentin is increased in fibroblast from old donors [29].

In this study, we used a combination of biophysical approaches to assess how cell morphology and biophysical behaviour are altered due to ageing. Human dermal fibroblast from donors of different ages were used as a model to characterise how changes in cell motility and biophysical properties are associated to changes in cytoskeleton organisation. Fibroblast from older donors had reduced cell motility and increased cell stiffness, which was associated to changes in cytoskeletal assembly. In particular the age-associated aberrations in cell motility and biophysical properties appeared alongside vimentin accumulation and could be rescued using drugs believed to primarily target vimentin. Our findings suggest the importance of vimentin in donor ageing and point towards this cytoskeletal protein and associated signalling pathways as potential biomarkers for the diagnosis, prognosis, and treatment of a wide variety of different diseases associated with ageing.

## 2. Materials and Methods

### 2.1. Cell Lines and Culture

Human dermal fibroblasts were obtained from commercial sources. In brief, cells were derived from temple or labia tissue from ”apparently healthy” Caucasian female donors. Neonatal (N) and adult age 62 (A62) cells were purchased from (Lonza Biologics, Slough, UK) while adult age 21 (A21) and age 47 (A47) cells were purchased from (PromoCell, Heidelberg, Germany). Vials of cells were shipped at passage 2, and all experiments were carried out in cells up to passage 7. NIH 3T3 cells were a gift from A.Mata group (Queen Mary University of London). All cell work was conducted in identical conditions among all donors, and culturing of cells was carried out in parallel. Cells were cultured on plastic plates in high glucose (4.5 g/L) DMEM medium (Thermofisher Scientific, Paisley, UK) supplemented with 10% foetal bovine serum (FBS) (Sigma, Poole, UK) and 1% penicillin/streptomycin.

### 2.2. Cell Transfection

Cells were transfected with actin (pCAG-mGFP-Actin, (21948) or vimentin (pVimetin-PSmOrange, (31922), (AddGene, Cambridge, MA, USA) plasmids. Cells were seeded at low density (2000–5000 cells/cm^2^) onto 6-well tissue culture treated plates in antibiotic free medium and allowed to adhere overnight. After this, cells were transfected with plasmids using a specific dermal fibroblast transfection reagent (Cambio, Cambridge, UK). The concentrations of plasmids and reagent were scaled down according to the number of cells per well. The transfection was allowed for 6 hours and after the fresh antibiotic free medium was replaced. All live experiments with transfected cells were performed 48 h after transfection.

### 2.3. Cell Migration

Transfected cells were seeded onto 6-well plates at low density. Prior to imaging, the medium was replaced with FBS supplemented Flurobrite-DMEM imaging specific medium (Thermofisher Scientific, Paisley, UK) to reduce background fluorescence and photobleaching. Time-lapse recordings of single cell dynamics were acquired with a 20× objective by a Lumascope LS720 microscope (Etaluma, San Diego, CA, USA) at a rate of 1 image every 10 min for at least 6 h. The miniaturised microscope was placed inside the incubator, so temperature and CO2 concentration were maintained throughout the time-lapse experiment.

For wound healing assays, cells were seeded at 25,000 cells/cm^2^ on 12-well plates with attached PDMS stencils and incubated for 2 days to confluence. The “wound” was initiated by removing the PDMS stencil, and the medium was aspirated and changed with fresh one. Cell migration was monitored by taking images every 30 min for 100 h.

The algorithm to analyse time-lapse fluorescence videos is based on grey-scale images of the fluorescent channels, and there are two steps: (1) determination of the cell outlines for every frame and (2) calculation of the positions of cell centroids. Once the position of the cell’s centroid was determined for each frame, we computed the cell’s instantaneous migration speed and the persistence of the overall recorded migration path as previously described elsewhere [30]. In brief, migration persistence is defined as the ratio between net cell displacement (the Euclidian distance between starting and ending centroid positions) and the overall distance travelled by the cell, as $P=\frac{d\left(X_{t\;=\;0},X_{t\;=\;T}\right)}{\sum_{i\;=\;0}^{T}d\left(X_{t\;=\;i},\;X_{t\;=\;i\;+\;1}\right)}$. Persistence values are thus unitless and bound between 0 (random migration) and 1 (straight line). For scratch assay experiments, the wound healing area (area not covered by cells) was calculated at 0, 24, 48, 75, and 99 h using ImageJ software (NIH, Bethesda, MD, USA). 

### 2.4. Cell Viscoelastic Properties with Atomic Force Microscopy

Atomic force microscopy (AFM) was employed to investigate the biophysical properties of human dermal fibroblasts. The AFM system (NanoWizard4, JPK, Berlin, Germany) was mounted on an epifluorescence microscope (Axio Observer Z.1, Zeiss, Jena, Germany). Images of live, healthy fibroblasts were scanned under liquid conditions (DMEM medium with 25 mM Hepes supplemented with 10% FBS and 1% penicillin/streptomycin) at 37 °C with the V-shaped gold-coated silicon nitride cantilevers (Budget Sensors, Sofia, Bulgaria) with four-sided pyramidal tips in contact mode. The cantilever had a spring constant of 0.06 N/m, length of 200 µm, and width of 30 µm. The spring constant of the cantilever was calibrated using the thermal fluctuations method based on sensitivity calculation on the bare region of the substrate. Force maps of the cells were taken in quantitative imaging (QI) mode at a resolution of 32 × 32 pixels, using 4000 nm ramp length, 250 µ/s ramp speed and a force setpoint of 2 nN. Using these conditions, maximum indentation levels reached were ~2 µm, typically on the vicinity of the nucleus of the softest cells probed. The scan area depended on the cell size, with the maximum attainable range being 100 × 100 µm^2^. If the cell exceeded that range, half or a quarter of cell was chosen including always a portion of the cell nucleus as well as the cell edge (Appendix A).

Biophysical properties such as Young’s modulus (E), viscosity (η), and non-specific adhesion work were determined from the force–distance curves. The force–distance curves were analysed using the BECC model for thin adherent cells on a stiff substrate [31], using a pipeline written in MATLAB as previously described [32]. Determination of Young’s modulus for the cell cytoskeleton (E_CSK_) and the cell cortex (E_cort_) was based on the approach proposed in Pogoda et al. [33]. In particular, and after the contact point has been identified, E_cort_ is obtained by fitting the force–indentation curve for data points corresponding to indentations <400 nm, whereas E_csk_ is obtained by fitting the force–indentation curve for data points corresponding to indentations >750 nm. Cellular viscosity was determined from force–distance curves using the method described by Rebelo et al. [34].

### 2.5. Immunofluorescence Staining and Imaging

Dermal fibroblast samples were prepared by seeding cells at low density (5000 cells/cm^2^) on 13 mm glass coverslips. The coverslips were coated with type I collagen at 10 µg/mL concentration for 1 h at 37 °C. After being rinsed with PBS, cells were seeded and allowed to adhere for 24 h. For drug treatment experiments, cells were seeded at the low density 24 h prior drug treatment. Cells were treated with 1 µM, 2.5 µM, and 5 µM concentration of withaferin A and with 2 mM, 4 mM and 6 mM concentrations of acrylamide for 3 h. Then, cells were fixed with 4% paraformaldehyde (Sigma, Poole, UK) for 20 min, washed with PBS, permeabilised with 0.25% Triton X-100 (Sigma, Poole, UK) for 10 min, washed with PBS, and blocked with PBS supplemented with 3% bovine serum albumin (Sigma) for 1 h at room temperature.

For the experiments of cell morphology, cytoskeletons, p21, alpha-smooth muscle actin, and nuclear organization, F-actin filaments were stained with TRITC-tagged phalloidin (1:1000, Sigma, Poole, UK) and co-stained with vimentin monoclonal mouse (1:300, RV202, Santa Cruz, CA, USA), tubulin monoclonal rabbit (1:200, ab4074, Abcam, Cambridge, UK), or yes-associated protein (YAP) monoclonal mouse (1:200, sc-101199, Santa Cruz, CA, USA), p21 monoclonal rabbit (1:250, 2947, Cell Signaling Technology, Leiden. Netherlands) or alpha-smooth muscle actin monoclonal mouse (1:200, A2547, Sigma, Poole, UK). Subsequently the secondary antibodies were used—anti-mouse Alexa 488 (A21202, Thermofisher Scientific, Paisley, UK) and anti-rabbit Alexa 488 (A21206, Thermofisher Scientific, Paisley, UK). After staining, coverslips were mounted on glass slides with Prolong Gold antifade mounting medium with DAPI (Thermofisher Scientific, Paisley, UK) to protect samples from drying out. Fluorescence images of the fixed cells were obtained using an inverted epifluorescence microscope (Leica DMI4000B, Wetzlar, Germany) with ×20/0.5 NA objective lens and a charged coupled device (CCD) camera (Leica DFC300FX, Wetzlar, Germany). Images were taken only on well-attached and not damaged cells using DAPI, FITC, and TRITC channels.

### 2.6. Single Cell Cytoskeleton Quantification Analysis

Our pipeline for single-cell quantification of cytoskeleton and nuclear structures has been described in detail elsewhere [30]. Briefly, the algorithm uses grey-scale fluorescence immunostaining-based or live-cell images typically obtained on epifluorescence or confocal microscopes, and it follows three independent steps: (1) initial fibre segmentation, (2) fibre refinement, and (3) determination and subtraction of non-uniform background within the cell boundaries. The algorithm outputs data at the single cell level, including gross cell morphology information like cell area, aspect ratio, and stellate factor or cytoskeleton information like fibre intensity, length, and thickness (for detailed descriptions and examples see [30]). To estimate fibre thickness in arbitrary units (AU), we measured the average pixel intensities for all pixels identified by the algorithm as belonging to a fibre. We note that in our imaging conditions, the pixel size is larger than the diffraction limit or the thickness of a single cytoskeletal filament. Accordingly, the measurement of fluorescence pixel intensity constitutes a good surrogate measure to estimate the number of individual fluorophores bound to a fibre and thus number of filaments making up a stress fibre or bundle. To estimate fibre length in microns, we computed the average length of the identified stress fibres or filaments in a cell in pixels and converted them to microns using previously-measured calibration factors matching the imaging conditions used. For nuclear data, the pipeline uses the DAPI images and provide estimates on the relative volume (compared to non-adherent conditions), chromatin condensation or Poisson’s ratio. In particular, the algorithm assumes that the gross morphology of the nucleus can be described as an ellipsoid, and uses changes in fluorescence pixel intensity along the radial direction of the nucleus to estimate the dimensions of its 3 semi-axes (for detail see [35]). Note only some of all the parameters output by the pipeline are used in this manuscript, corresponding to their relevance to the present research question.

### 2.7. Cell Reattachment Experiments

The reattachment experiments were carried out using the Lumascope LS720 (Etaluma, San Diego, CA, USA) microscope as above, using only healthy and well-attached transfected cells. To initiate the reattachment event, cells were treated with trypsin until they displayed a rounded up shape but before they were completely detached. Subsequently, fresh imaging medium was added to the wells and the process of cell reattachment was imaged. Fluorescence images were recorded every 10 min for 6 h using a 20× objective.

### 2.8. Drug Treatments Against Vimentin

Healthy vimentin-transfected and well attached cells were chosen and imaged for 1 h prior drug treatment. Subsequently, Withaferin A (Sigma, Poole, UK) with concentrations of 1 µM, 2.5 µM, and 5 µM, or acrylamide (Bio-Rad Laboratories, Watford, UK) with concentrations of 2 mM, 4 mM, and 6 mM was added and cells were imaged for 6 additional hours. Images were captured every 10 min. Cell velocity was calculated as described above on the same cells before and after drug treatment.

### 2.9. Statistical Analysis

Statistical analysis was performed using GraphPad Prism 5 software (GraphPad Software, San Diego, CA, USA). The *t*-test was used for the normally distributed data sets, otherwise, the non-parametric Mann–Whitney *U* test was adopted. Statistical significance was reported at *p* < 0.05 (*), *p* < 0.01 (**), and *p* < 0.001 (***) unless otherwise stated. All experiments were performed using at least three replicates unless otherwise mentioned in the figure legend.

## 3. Results

### 3.1. Donor Age Reduces Cell Migration and Increases Young’s Modulus of Human Dermal Fibroblasts

The purpose of this study was to evaluate the biophysical properties of human dermal fibroblast cells obtained from donors of different ages, obtained at ages: Neonatal, 21, 47, and 62 years. To measure the cell velocity of single cells, a miniaturised live imaging system placed inside an incubator was used to perform long-term cell migration experiments in 2D at physiological conditions. Cells were seeded at low density onto six-well plates and transfected separately with a fluorescently-tagged vimentin plasmid. Transfected cells were allowed to recover for 48 h prior to migration experiments. Images were taken only of single cells that were clearly transfected, healthy, and well attached. Time-lapse fluorescence images were taken every 10 min for 6 h. The videos of cell migration were then analysed to measure migration velocity and directionality, by tracking the non-fluorescent circular area corresponding to the cell nucleus. The results show that human dermal fibroblast cells from the neonatal donor have a significantly higher velocity compared to all adult donors. The largest difference (twofold) was observed when comparing them to cells from the oldest donor (Figure 1A). Interestingly, cell persistence was affected only when comparing cells from the neonatal to the oldest donor (Figure 1B). Scratch assays yielded similar trends, with the oldest donor showing delayed migration into the scratch, even though no differences were observed for the other donors (Appendix A). Of note, the rate at which the wound closes is affected by the migration speed of cells but also by the average spread area of the cells. Given that both are affected by donor age, our results measuring individual cell migration thus constitute a less incumbered method and provide clearer results. To rule out that the observed differences in cell migration were not due to other differences between the primary cells used, we quantified nuclear expression of p21, as a marker of cell proliferation, and cytoplasmic expression of α-smooth muscle actin (α-SMA), as a marker of myogenic differentiation. In both cases, we did not observe clear trends with donor age or cell spread area but found a slight but significant increase on p21 nuclear expression for the A62 donor (Appendix A) and a slight but significant decrease in α-SMA for the A47 donor (Appendix A). Altogether our results suggest that donor age has a significant impact on cell motility, which may delay the capacity of dermal fibroblasts to engage in wound healing.

Cell motility is associated with changes in biophysical properties, which are regulated by the cytoskeleton. We therefore examined whether donor age has an effect on cell biophysical properties using atomic force microscopy to measure viscoelastic properties. Individual cells from all groups were probed in QI mode, and our customised data-analysis pipeline was used to calculate cells’ Young’s modulus (E), viscosity, and adhesion work. When determining E, we found that cells from the oldest donor displayed a twofold increase compared to cells from the neonatal donor (Figure 1C). Similarly, the measurement of cell viscosity showed a significant 1.4-fold increase for cells from adult donors compared to cells from the neonatal donor (Figure 1D). Furthermore, when evaluating cell adhesion work, we found significant differences also between cells from the neonatal donor compared to cells from the oldest donor, the increase being 1.5-fold (Figure 1E). While previous studies using immunostaining have demonstrated that adhesion proteins increase in senescent cells [36], it is worth pointing out that we report here unspecific adhesion values, given they were determined as adhesion strength between the cell membrane and untreated silicon nitride cantilevers tips.

Together, our results show that donor age significantly affects biophysical properties, and in particular induces a reduction in cell motility alongside increased cell elastic modulus, viscosity, and adhesion force.

### 3.2. Cellular and Nuclear Morphology of Human Dermal Fibroblasts Depend on Donor Age

Changes in donor aging have been linked to alterations in cellular morphology [37], and hence we examined whether the observed aberrations in migration and mechanical properties of human dermal fibroblasts from older donors were associated with changes on their underlying cytoskeleton. First, cellular and nuclear morphology was quantified from epifluorescence images of cells labelled with phalloidin for F-actin and DAPI for the nucleus (Figure 2A). Human dermal fibroblasts from older donors displayed a significant increase in cell area compared to cells from the neonatal donor cell surface area, which was around 2000 µm^2^ (Coefficient of Variance, CoV = 58%) for cells for the neonatal donor, while for cells from adult donors, the surface area was larger and ranging from 3000–7000 µm^2^ (CoV = 60%), reaching a larger than twofold increase when comparing cells from neonatal donor to cells from oldest donor (Figure 2B). With increasing donor age, cells also underwent changes in their aspect ratio, from a spindle shape to large solid spread (Figure 2C). Interestingly, the changes in cellular morphology and specifically the increases in cell spread area had only a weak correlation with changes in nuclear volume. In this regard, the nucleus volume increased significantly only when comparing cells from the youngest to the oldest donors (Figure 2D).

Previous studies have suggested that YAP localization is regulated by cell-matrix interactions and intracellular tension during cell attachment and spreading [38]. Since our results showed age-associated changes in cell biophysical properties and specifically in cell spreading area, we examined whether they would lead to changes in YAP intracellular localization. We cultured cells at low density and labelled them with phalloidin for F-actin, YAP primary antibody and DAPI for cell nucleus. In this experiment, phalloidin staining was used to readily quantify cell area, and we used imaging protocols as described above. To measure YAP localisation, specifically whether YAP is localised preferentially in cell nucleus or cytosol, we quantified YAP nuclear to cytosolic ratio as done by others [39,40]. Representative fluorescence images show that YAP is more concentrated in the cell nucleus in cells from the neonatal donor compared to cells from older donors (Figure 2E). In particular, cells from the oldest donor show a 1.6-fold reduction in YAP ratio compared to cells from the neonatal donor (Figure 2F). We next verified whether there was any connection between YAP localisation and cell area and found that increasing cell areas lead to decreased YAP ratios in a strongly correlated manner. Surprisingly, the relationship between YAP localization and donor age appeared to be only secondary, as shown by the strong overlap between data points for all donor ages in Figure 2G. These results suggest that changes in YAP ratio are primarily associated with changes in cell area, which is on its own regulated by donor age.

### 3.3. Vimentin Rather Than F-Actin or Microtubules is Dominantly Increased in Human Dermal Fibroblast Ageing

The three main cytoskeletons, F-actin, microtubules, and the intermediate filament vimentin are all key players in maintaining cell morphological and biophysical properties. Since our results indicated that donor ageing modulated cell biophysical properties and morphology, we next investigated whether this was associated with changes in F-actin, tubulin, and the intermediate filament vimentin. Similar to previous immunostaining experiments, cells were cultured at low density and then stained with phalloidin for F-actin and primary antibodies against tubulin or vimentin. Single cells were imaged using epifluorescence microscope equipped with 20× objective. Quantification algorithms were used to determine cell morphology as well as properties of fibre architecture and overall organisation. Representative images show that cells from aged donors had more pronounced actin fibres compared to cells from the neonatal donor (Figure 2A) as well as similar changes for tubulin and vimentin fibres (Figure 3A). In particular we found a significant increase in F-actin amount alongside a significant decrease in actin fibre length and thickness in cells from aged donors compared to cells from neonatal donor (Appendix A). Donor age had an effect not only on F-actin but also on vimentin fibre morphology. The results indicate that cells from older donors have an increased amount of vimentin with longer and thicker fibres (Appendix A). Similarly, cells from older donors showed increased levels of tubulin amount with shorter and thicker fibres compared to cells from the neonatal donor (Appendix A). Together, these data show that donor aging is associated with changes in all three cytoskeletons. We then normalised our cytoskeletal amount data to account for differences in primary and secondary antibody affinities that lead to dissimilar amounts of fluorescence intensities being measured for each stained cytoskeletal protein. When reporting relative changes against the measured cytoskeletal amount of the neonatal donor, we found that vimentin displayed the largest increase with donor age (Figure 3B–D).

Accordingly, we decided to further focus on the intermediate filament vimentin and explore its dynamics. To study the dynamics of vimentin fibres in live cells, we developed a single cell reattachment experiment as follows. Cells at low density were initially transfected with vimentin plasmid and treated with trypsin for a short period of time until they displayed a rounded up morphology without being completely detached. Immediately afterwards, trypsin was gently exchanged with fresh medium, and selected cells were imaged with a 20× objective. Images were taken every 10 min for 10 h. During the reattachment process, changes in cell area and vimentin fibre dynamics were clearly observed (Appendix A). We then investigated whether vimentin fibre remodelling rate during reattachment was affected by donor age. To do so, vimentin fibre amount was quantified for all the frames in the videos obtained during the reattachment process. The representative plots of vimentin fibre versus time show that the amount of vimentin reaches a plateau, whose value increases in cells from older donors (Figure 4A), in a fashion similar to the results obtained for immunostaining. To extract additional information about reorganization dynamics, we fitted our data using a one-phase exponential decay function: $y=y_{0}+\left(plateau-y_{0}\right)\cdot \left(1-\exp\left(-k\cdot x\right)\right)$. From the fitted data, we derived parameters such as half-life, computed as 1/k; or span, computed as $\left(plateau-y_{0}\right)$. The half-life parameter estimates the dynamics of vimentin during reattachment, while span estimates the amount of vimentin once the cell has established full reattachment (Figure 4B). Our results show that vimentin fibre remodelling rate is faster for neonatal cells (smaller values for half-life) and decreases with donor age (Figure 4C). The span results again agree with immunofluorescence data, which show increased vimentin steady-state amount in older cells (Figure 4D).

### 3.4. Drug-Induced Changes in Vimentin Assembly are Correlated with Changes in Cell Motility and Young’s Modulus

Given that cells from older donors displayed reduced motility and an increased number of vimentin fibres, we next explored whether biophysical properties of cells could be modulated using drugs believed to primarily affect vimentin assembly. To do so, we used withaferin A and acrylamide and monitored single cell migration after treatments with said drugs in neonatal and adult cells (using 47-year-old donor source). Cells at low density were transfected with vimentin-GFP for 72 h prior to drug treatment, and time lapse fluorescence images were taken only on transfected and well-attached cells. Considering the large variability of single-cell motility, we decided to image the same individual cells before and after drug treatment. Therefore, cells were imaged for 1 h before treatment and 3 h after drug treatment. Cell velocity was measured as previously described. In parallel, a different set of cells treated with the same drugs were immunostained with phalloidin for F-actin and primary antibody against vimentin to quantify their assembly.

First, we investigated the potential effect of withaferin A treatment on cell migration and vimentin assembly. Withaferin A treatment caused a reduction in cell motility and increased the amount of vimentin assembled in fibres for cells from the neonatal donor (Appendix A). Similar results in terms of cell motility and vimentin assembly were observed in cells from the older donor (Appendix A). Of note withaferin A treatment caused aggregation of vimentin fibres, which was already observed in previous study [41] as well.

Next, we investigated the effect of acrylamide treatment using the same approach as before. Surprisingly, acrylamide treatment had no effect on cell migration of cells from neonatal donor and showed a minor effect on vimentin and F-actin fibres assembly (Appendix A). However, a significant increase in cell motility alongside a significant reduction in vimentin fibres was found for cells from the aged donor (Appendix A). These results suggest that withaferin A and acrylamide have an opposite effect on vimentin assembly in our cells, which is partially dependent on donor age. Therefore, we pooled all results together from withaferin A and acrylamide treatments using only the two highest dosages. Surprisingly, we found a strong correlation between relative changes in cell velocity and relative changes in vimentin amount due to drug treatments. In particular, withaferin A caused a reduction of cell velocity and increased vimentin amount; meanwhile acrylamide treatment increased cell velocity and reduced vimentin amount (Figure 5A). To confirm that this effect was primarily associated with changes in vimentin fibres, we verified that there was no correlation between cell velocity and F-actin relative changes with either withaferin A or acrylamide treatments (Figure 5B).

Since withaferin A and acrylamide had a modulatory effect on vimentin fibres, which was observed alongside changes in cell motility, we next investigated whether a similar correlation was observed for Young’s modulus (E) and whether those effects depended on the mechanical structure being probed. Atomic force microscopy was employed to measure the viscoelastic properties of cells treated with withaferin A or acrylamide. Force indentation–curves were taken by probing cells treated with withaferin A or acrylamide after 3 h. We found that withaferin A treatment increased E_CSK_ and E_cort_ of cells from the neonatal donor (Appendix A), even though significant differences were only observed for E_cort_ at the highest concentration. Conversely, withaferin A treatment had no effect on cells from the adult donor (Appendix A), suggesting that it was unable to further stiffen the already reinforced cytoskeleton of old cells. Similarly, we investigated the effect of acrylamide treatment on E and found that cells from both neonatal and adult donors showed a significant reduction in E_CSK_ (Appendix A), while E_cort_ was not affected for both donor ages (Appendix A). This result is not surprising given that the vimentin network is primarily localised deep in the cell body, whereas actin is the mechanically dominant structure in the cell cortex.

Together, these results suggest that there is a significant correlation between E_CSK_ and vimentin assembly (Figure 6A,B), which parallels the correlation between cell migration speed and vimentin assembly. While we find that withaferin A and acrylamide treatments had a mild effect on actin assembly, this was not correlated with E_CSK_, E_cort_ (Figure 6C,D) or cell migration speed. 

Accordingly, our findings indicate that for human dermal fibroblasts, cell biophysical properties such cell motility and Young’s modulus are primarily correlated with amounts of vimentin assembled in filaments. Specifically, treatments on older cells that lower the amount of vimentin to levels comparable to those displayed by younger cells also result in the rejuvenation of the biophysical and migratory phenotype displayed by older cells.

## 4. Discussion

In this study, human dermal fibroblast cells from donors of different ages were used as a model to study how single cell migration, biophysical, and morphological properties are altered by donor age. In recent years, a number of studies have focused on characterizing delays in wound healing associated with cellular aging [42,43]. In particular, cell velocity is considered a key biophysical parameter, which is widely used to characterise the cell’s ability to move from a healthy to a diseased location within its host tissue [44,45,46,47]. Previous studies have focused on proteasome content and activity to understand cell senescence [48], but little is known on how cell biophysical and morphological properties are associated with donor age. Here, we show that donor aging resulted in reduction of cell motility, which was associated with cell stiffening and increased amounts of F-actin, tubulin, and dominantly vimentin.

The cytoskeleton is a complex system with a broad range of functions, such as the formation and maintenance of cell morphology, polarity, cell division, and migration. Cells from aged donors displayed changes in cell morphology with a reduction in cell motility and increased mechanical strength. It is thus expected that the integrity of the cytoskeleton is altered, not only at the macrostructure but also at the nanostructure level. F-actin fibres are believed to be key factors in regulating cell shape and motility, although microtubules and intermediate filaments play a crucial role too. In this connection, changes of F-actin structure and amount have been reported in cells undergoing induced senescence. For one study, cells had thicker fibres but the total amount of F-actin remained the same [36]. Meanwhile in another study, the total amount of actin protein was observed to be reduced in cells from aged donors [49]. Among other cytoskeletal networks, changes of the intermediate filament vimentin have been reported in several types of senescent cells. Using extensive passage as a surrogate for cellular aging, vimentin was found to develop thick and long fibres, while cells at early passage had thin and short fibres [50]. Similarly, it has been reported that the number of tubulin fibres also increases in senescent cells [37]. In this study, we report for the first time that all three cytoskeletons are altered by donor age. F-actin, tubulin, and vimentin all increased in abundance for cells from adult donors, displaying shorter and thinner fibres for F-actin and tubulin and longer and thicker fibres for vimentin. Focusing on vimentin as the most reinforced structure, we found that vimentin fibre remodelling rate is slower, with higher level of protein in cells from adult donors. These changes suggest that the increased assembly of vimentin filaments observed in cells from older donors plays an important role in the aberrant biophysical properties associated with donor aging.

Yes-associated protein (YAP) has been shown to be regulated by cell senescence [51]. Here, we show that changes in YAP ratio are most likely primarily associated with changes in cellular gross morphology (Figure 2G). Therefore, YAP ratio changes are indirectly dependent on donor age, as cells from aged donors display larger spreading areas that leads to lower YAP ratios. On a different note, observations by others indicate that senescent cells have larger spread areas [52]. While on average, the population of cells from the A62 donor displayed a light increase in senescence (reduced proliferation) marker p21, we did not find correlation trends between nuclear expression of p21 and cell spread area when we performed our analysis on a single-cell basis. Put together, these suggest that the aberrations in biophysical parameters we observe for cells from older donors are likely linked to changes in vimentin assembly, rather than being linked to the onset of senescence. Of note, in a different unpublished study, we find that extensive passaging (more than 15 passages) of neonatal cells leads to similar biophysical properties to those displayed by early-passage cells from older donors. Conversely, extensive passaging of cells from older donors does not result in further reinforcement of the cytoskeleton and cell mechanics, but rather leads to an aberrant mechanical phenotype that may represent the onset of senescence.

Vimentin has been known to play a key role in cell migration. In migrating fibroblasts, the nucleus is surrounded by an abundance of vimentin filaments, which extend into the tail of the cell. On the contrary, vimentin monomers and short filaments are localised at the leading edge. These intracellular regional changes in vimentin structure and organization are responsible for regulating protrusion activity. In addition, serum starvation in fibroblasts caused reduced motility and the local breakdown of the vimentin network [53]. Similarly, vimentin assembly is essential for wound healing in several animal models and cells in culture [25,26,54]. Fibroblasts from vimentin-deficient mouse exhibited a reduction in cell motility, defects in directionality and on their ability to organise collagen [25,55], while vimentin overexpression caused increased cell motility of breast cancer cells [56]. These findings indicate that vimentin filaments play an important role not only in cell mechanical support but also in cell motility and that an exquisite fine tuning of its amount and organization is required for optimal cell migration.

The contribution of vimentin organization to cell motility and mechanical properties can be also assessed using drugs against vimentin. Of note, the use of drugs targeting the polymerization of vimentin monomers into filaments rather than the use of siRNA against vimentin protein expression is an approach that parallels the use of cytochalasin D or latrunculin A against the assembly of G-actin monomers into F-actin fibres to understand the structural role of stress fibres on cell mechanics. That being said, the existing biochemical toolkit to target vimentin is still very limited and not fully characterised. Accordingly, the two gold-standard drugs used in the literature, withaferin A and acrylamide, may also affect other cytoskeletal structures or signalling pathways in addition to modulating vimentin filament assembly. In our experiments, withaferin A treatment induced disruption of vimentin organization and led to the formation of aggregates, which are believed to be associated with changes in cell shape, reduction in cell motility [57], and cell softening [41]. Similarly, cells treated with acrylamide have been reported to display reduced stiffness, as evaluated by applying large strains on cells embedded in alginate gels [58]. Interestingly, in our study, we find that cells from both neonatal and adult donors treated with withaferin A at concentrations of 1–5 µM displayed reduced cell motility and increased cell stiffness which was likely associated with aggregation of vimentin. We thus hypothesise that the observed cell stiffening is associated with changes in vimentin organization from long filaments to short structures and aggregates. The aggregates then formed solid, stiff structures, which increased cell stiffness. Furthermore, we find that withaferin A-associated changes in vimentin organization, cell motility, and Young’s modulus are dose- and donor-age dependent. In contrast, cells treated with acrylamide exhibited increased cell motility and reduced Young’s modulus, which was correlated with a reduction of vimentin assembly. We and others have shown that the modulation in the assembly of stress fibres, microtubules, and intermediate filaments is often analogous and closely tied to cell spread area [30,35]. It is thus plausible that the drug treatments against vimentin used here did also induce changes in the assembly of other cytoskeletal filaments. Nevertheless, in our experiments, we used shorter treatments and lower concentrations than those used by others when reporting detrimental effects of these drugs on all cytoskeletons [59,60,61]. Similarly, the strong correlation observed between vimentin assembly and biophysical properties was largely lost when we performed similar analysis using instead levels of actin filamentous assembly. Together, our results suggest that in the cellular model used here, vimentin assembly has a dominant role in modulating the biophysical and migratory behaviours. It is worth mentioning that experiments by others on vimentin knock-out cells show aberrant biophysical behaviours, with a significant decrease in cellular stiffness as well as migration speeds [24,62]. Accordingly, we hypothesise that the amount of vimentin fibrillar assembly, rather than overall level of vimentin protein expression, plays a crucial role in fine-tuning cell mechanics to attain optimal migration rates. It thus follows that a complete inhibition of vimentin assembly does not necessarily increase cell migration further and that a certain amount of vimentin is likely necessary for optimal cell motility.

In summary, and to highlight the relevance of our results, we show that vimentin dominates the changes in cytoskeleton organization and assembly in human dermal fibroblast cells and may thus play a key role in the aberrant behaviour and impaired function displayed by this cell type in the course of human ageing. Accordingly, we propose that vimentin might serve as a suitable therapeutic target especially for aging-related diseases. We further propose that biophysical properties such cell motility and mechanical properties are strongly correlated to vimentin amount and can thus be readily used as high-throughput biomarkers on drug screening assays in the search for new anti-aging therapies.

## Figures and Tables

**Figure 1 cells-08-01164-f001:**
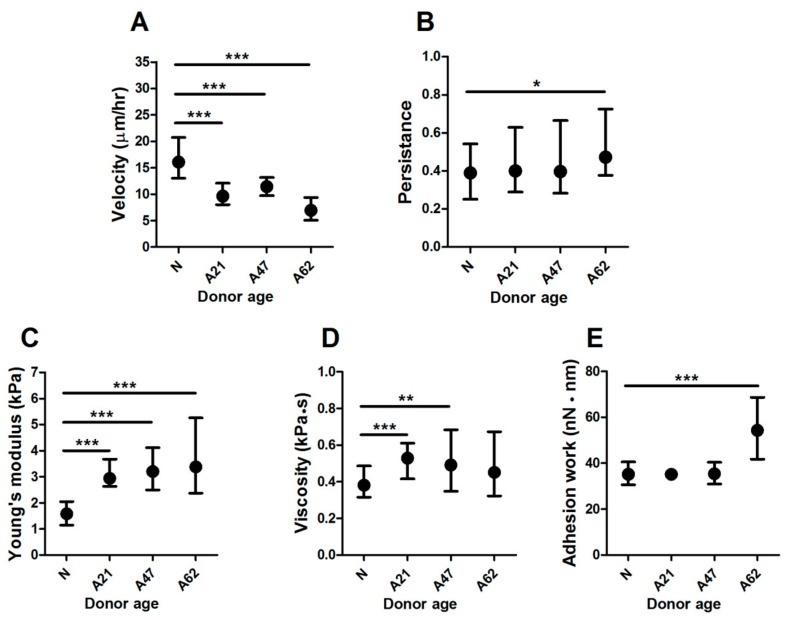
Biophysical properties are altered by donor age. (**A**) Corresponding plot showing reduced cell velocity of single fibroblasts on two-dimensional substrates in relation to donor age. Cell persistence was significantly different only for cells from oldest donor (**B**). Data plotted from at least three independent experiments as geometric mean with quartiles, cell number varies between (50–60). Cells from aged donors exhibited increased viscoelastic properties compared to cells from neonatal donors as quantified by significant differences in (**C**) Young’s modulus, (**D**) viscosity, and (**E**) adhesion work estimated using AFM measurement. All data plotted from at least three independent experiments as geometric mean with quartiles, ** *p* < 0.01, *** *p* < 0.001, Mann–Whitney *U* test. Cell number varies between 30–90 with ~12 cells per repeat.

**Figure 2 cells-08-01164-f002:**
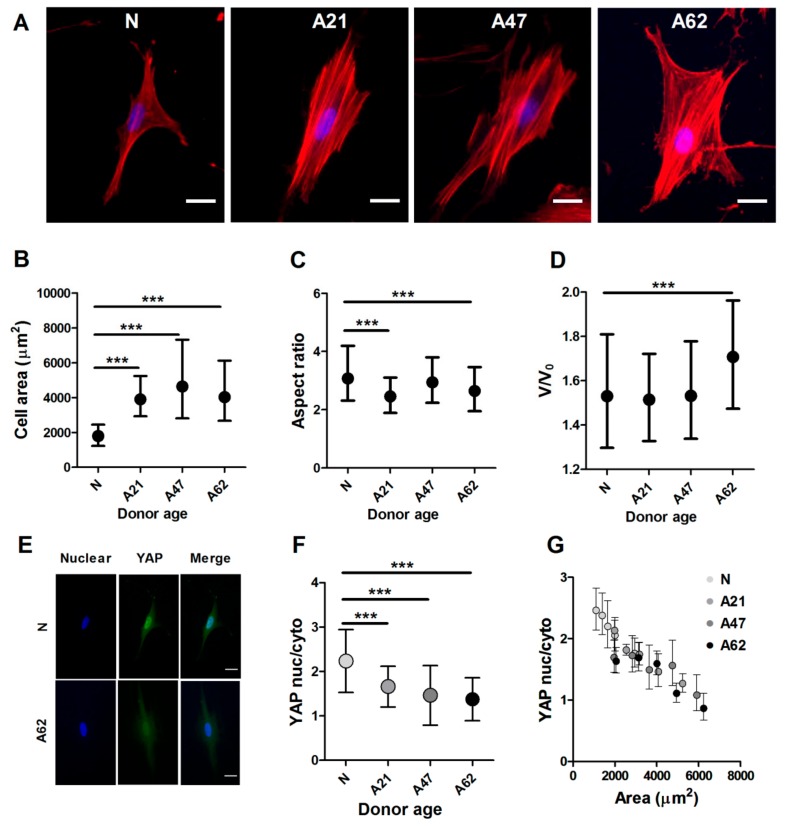
Cells from aged donors exhibited changes in cellular and nuclear morphology. (**A**) Represantative epifluorescence images showing increased cell area and F-actin organization of cells from donors at different ages. F-actin labelled with phalloidin (red), nucleus DAPI (blue). Scale bars represent 20 µm. (**B**) Corresponding plot showing the increased cell area of cells from aged donors compared to neonatal donor. (**C**) There was a significant decrease in aspect ration of cells from aged donors (*** *p* < 0.001, Mann–Whitney *U* test). (**D**) Nucleus relative volume increased for cells from aged donor (*** *p* < 0.001, Mann–Whitney *U* test). Data plotted from at least three independent experiments as geometric mean with quartiles, total cell number varied between 282–620, with ~100 cells per repeat. (**E**) Representative epifluorescence images of yes-associated protein (YAP) localisation in cells from neonatal and adult donors. Cells are labelled, nucleus DAPI (blue), YAP (green). Scale bars represent 20 µm. (**F**) Immunostaining analysis showing a significant reduction of ratio of YAP nuclear to cytoplasmic in cells from aged donors. (**G**) Corresponding plot showing correlation between YAP localisation to cell area. Independently to donor age, in all age groups larger cells have less nuclear YAP. Data are plotted from three independent experiments and presented as mean values with SD (nonparametric one-way ANOVA test, *** *p* < 0.001). Number of cells ranged between (68–202).

**Figure 3 cells-08-01164-f003:**
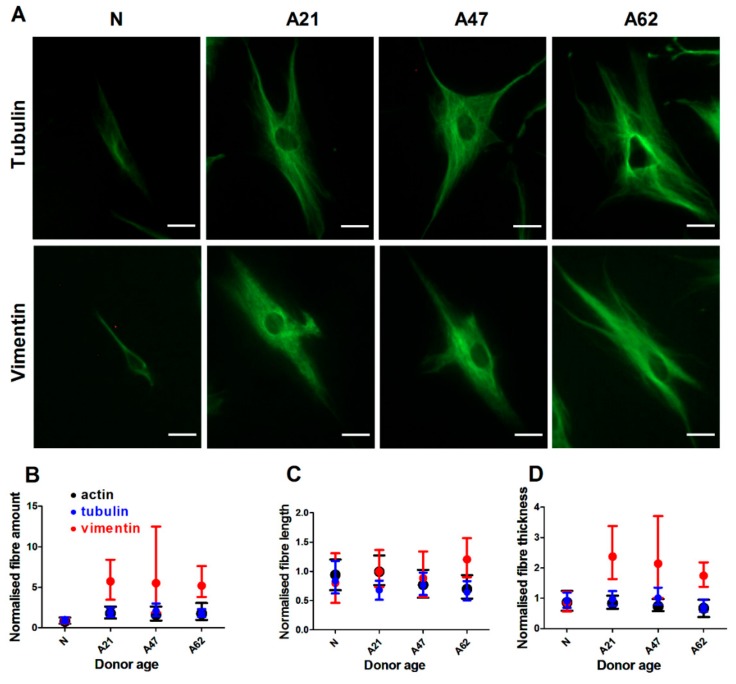
Age has the highest influence on intermediate filaments. (**A**) Represantative epifluorescence images showing tubulin and vimentin organization of cells from donors at different age. Fibres labelled ageinst tubulin and vimentin (green). Scale bars represent 20 µm. Corresponding plot of three cytoskeletons showing relative changes in (**B**) fibre amount, (**C**) fibre length, and (**D**) fibre thickness. Vimentin is showing the highest changes compared to F-actin and tubulin. Data are plotted from three independent experiments and normalised to neonatal donor to show the magnitute of changes. The data of all cytoskeletons with real values are presented in Appendix A.

**Figure 4 cells-08-01164-f004:**
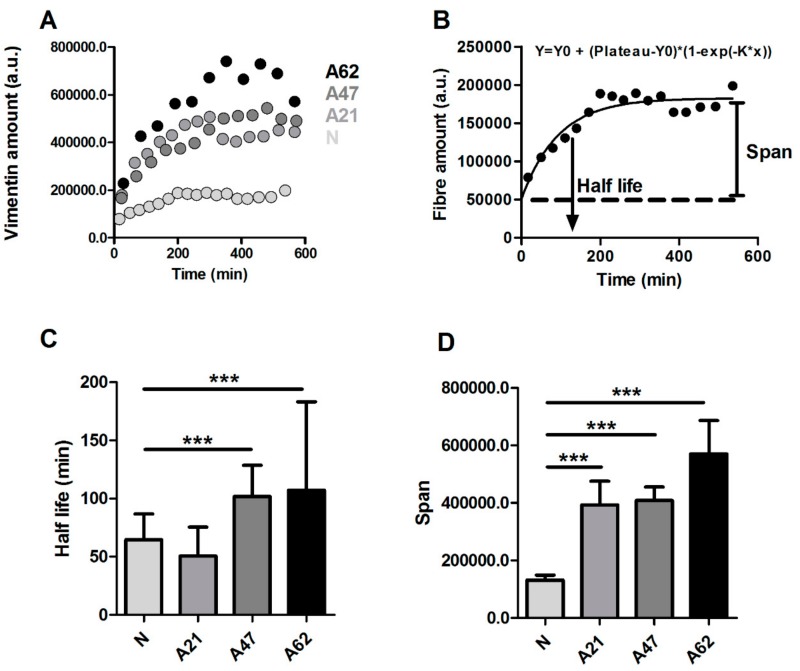
Vimentin fibre remodeling rate is faster in cells from young donor. (**A**) Representative plot shows the temporal changes in vimentin fibre intensity during reattachment time of cells from different donor ages. (**B**) An exponential model was used to fit the data and to determine parameters such as half-life and span. (**C**) There was a statistically significant difference in vimentin fibre remodeling half-life, indicating slower vimenitin remodelling rate for adult cells. (**D**) The span indicates that the vimentin fibre amount is higher in adult cells. Data are plotted from at least three independent experiments and presented as mean values with SD (*** *p* < 0.001, obtained using Dunnett test against neonatal donor). Cell number varies between (26–35).

**Figure 5 cells-08-01164-f005:**
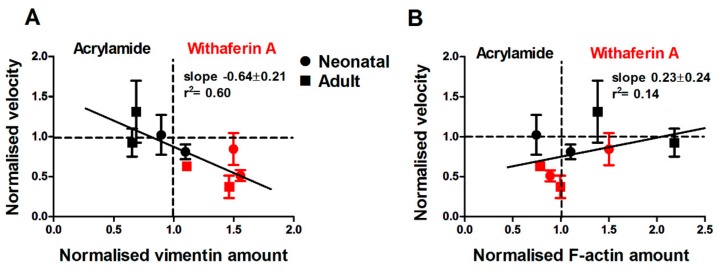
Acrylamide and withaferin A treatments have a reversable effect on dermal fibroblast migration and vimentin and actin fibres. (**A**) Corresponding plot showing the increased cell velocity of cells treated with acrylamide and decreased cell velosity of cells treated with withaferin A, which correlated with changed in vimentin amount. (**B**) The plot shows that cell velocity was not correlated to changes of F-actin amunt. Plots are presented using acrylamide 4 and 6 mM, and withaferin A 2.5 and 5 µM concentrations. The data of all concentrations are presented in Appendix A.

**Figure 6 cells-08-01164-f006:**
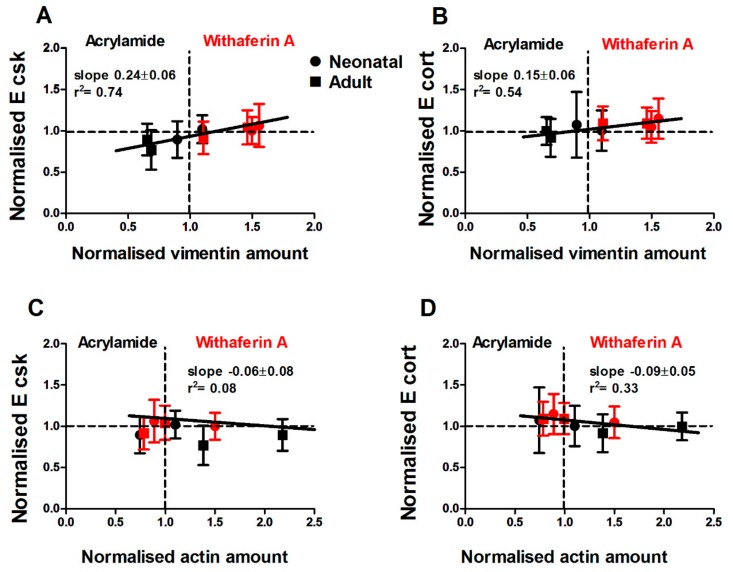
Changes in vimentin but not actin amount modulate E_CSK_ and E_cort_. Corresponding plots of (**A**) cytoskeleton and (**B**) cortical stifness show a significant correlation to vimentin amount. Cell treatments with withaferin A and acrylamide show a correlation between E and amount of vimentin. Changes in actin amount in treated cells do not effect cell (**C**) cytoskeletal and (**D**) cortical Young’s modulus. Plots are presented using acrylamide 4 and 6 mM, and withaferin A 2.5 and 5 µM concentrations. The data of drug treatment and significant differences are presented in Appendix A.

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
