# Peer review of "Vimentin Plays a Crucial Role in Fibroblast Ageing by Regulating Biophysical Properties and Cell Migration"

_cells, 2019, doi:10.3390/cells8101164_

Round 1

Reviewer 1 Report

      In this manuscript, Sliogeryte and Gavara investigate how biophysical properties differ between human dermal fibroblasts using donors of increasing age, and suggest that the intermediate filament protein vimentin is reponsible for the observed differences. Although the approaches are valid and experiments carefully designed and performed, and the research question interesting there are a number of concerns with the manuscript:

 - to the M&M section, I suggest that the authors add detailed information about the cells used ( Neonatal (N) and adult age 62 (A62) (Lonza 75 Biologics), adult age 21 (A21) and age 47 (A47) (PromoCell). Are these cells comparable, and all male/female/caucasian/healthy and from the same tissue, and subtype of fibroblast? Have they all been cultured on tissue plastic/glass for the same number of passages or the same amount of time? How did the authors characterize the cells, do some cell types show myofibroblast markers?

- I would also suggest that the authors add a section in the main text, where they explain how they can exclude the possibility that the observed differences are due to other factors. I would add a section discussing if the age difference is the only difference between these cells, or could the differences be due to other factors? If they do not exclude this possibility, then I suggest that they discuss this in the discussion section of the manuscript.

- In order to exclude the possibility that the observations are due to other differences between the cells, I suggest that the authors validate the observations in an isogenically matched cell model or aging. They can, for example, induce senescence in normal fibroblasts, compare normal to senescent cells, to see if they can observe the same differences.

- To the introduction, I suggest adding this finding and reference: "Berdyyeva TK et al. (2005). Human epithelial cells increase their rigidity with aging in vitro: direct measurements. Phys. Med. Biol."

-  Withaferin A and acrylamide cannot be considered to target only vimentin. To my opinion, any claims regarding vimentin causing the observed effects would need to be shown using vimentin-targeting siRNA. 

Reviewer 2 Report

Author's theory that age-related deteriolation is caused by cellullar disfunction is not supported by literature.

Only 4 human cell lines were investigated. And there is no data of humal cell over 70 years-old.

Reviewer 3 Report

The authors addressed interesting questions regarding the nanomechanical properties of fibroblast cell lines provenient of a series donors from neonatal to aged ones. Although a bit more care should be taken for the authors not to confuse in the text of the manuscript the age of donors with the age of the studied cells, the presented results might be of great interest. The authors show that the cytoskeletal networks, especially the intermediate filaments formed by vimentin might be dominantly responsible for age related behavior of the studied cells. I find this approach interesting, although a bit too simplified and in several cases the shown explanations lacking important details.

Questions & comments:

It is a bit confusing in the manuscript, that cellular age is sometimes mentioned as a synonym for donor age, however they are completely different parameters. Please clarify and rephrase adequately. (E.g. line 252, 344)

How specific is withaferin A and acrylamide to vimentin? Do they bind to other molecules? Are there any data for nanomechanical properties of vimentin lacking cells treated with withaferina A or acrylamide?

Did the authors compared their data to classical wound healing assays, where a scratched cell layer is monitored how to migrate and fill the open area?

Larger apparent area is a typical marker for senescent cell. Have the authors checked for senescence markers in case of those cells from aged donors?

Authors compare nuclear volume on Fig2D, but they do not mention how volume was calculated from 2D images? What is the scale of Fig 2D?

How was total vimentin amount and fiber amount calculated in case of Fig 4. Was it divided by apparent cell size? Otherwise this might be a simple size related increase, and half life and span are only cell motility parameters. Please clarify.

How exactly vimentin fiber remodeling rate was calculated?

During force curve recordings, what was the applied load? What indentations were achieved? While in manuscript adhesion force is stated, Fig 1 E shows adhesion work which is related to adhesion force but not identical. What is the scale of Fig 1E?

How cell persistence was calculated?

Cell numbers used in experiments are given as a number between a lower and higher value, do they stand for total number or for parallel single replicates of experiments? Since cell numbers differ largely were there any normalization to actual number introduced into presentation of the results?

Results for FigS1 come from 3T3 cells, which is an immortalized mouse cell line. How can this result apply for the used human fibroblasts?

What were the distinction criteria to calculate E csk and E cort? The cited reference (Ref 31) does not provide any information on this. Line 154 states stiffness while the whole manuscript deals with Elastic modulus. They are completely different quantities, please unify the nomenclature since usage as synonyms might be misleading.

Figure S2 panels A,D,G have scales of arbitrary units. Since all others lack specified scale units, is “a.u.” applicable to other panels in the named figure? If so, please note in caption or somewhere else in the text. Although the text does not states, were there any super-resolution techniques applied to quantify the length and thickness of cytoskeletal filaments? If not, how exactly the presented values were achieved?

The methods section states that QI maps of 32 x 32 resolution were recorded. Besides the reliability of the results, it would highly augment the level of the manuscript if (in case of size limits, at least in the supplementary material) some of these maps would be presented (as representative examples).

Panels of several figure lack scale units determinations. Please adjust them by labeling with the used units.

What exactly “dreased cell velocity” means? (Line 355)

Spelling errors (mainly in figure captions): amunt (line 357), cytoskeletol (line 381), vientin (line 384), ageinst (line 283), normalisd (line 286), velosity, canditions (Caption Fig S3). The list is not complete, just shows some examples which should be corrected before the final version is submitted.

Most of those requests referring to calculation or distinction methods should be briefly noted in the methods section, or at least in the supplementary material. Their exact details might ease the understanding and appreciation of the presented results.

Round 2

Reviewer 1 Report

The findings are interesting. 

However, in the present state, the data does not fully support the strong claims they make regarding vimentin functions.

In their answers to two of my questions, the Authors refer to unpublished findings that I do not have access to, or can evaluate, and these findings that are not included in the manuscript. Therefore, I do not consider that these unpublished observations can be used to answer my questions. 

The way I see it, my concerns regarding the cell types and anti-vimentin drug used are not fully addressed. I am aware of the technical limitations that the Authors face. It is difficult to obtain the matched cell types that I suggested, and specific anti-vimentin drugs do not exist. However, nonetheless, I do not consider that the data, in its present form, can be used to make the strong claims on vimentin function that the Authors make. 

Reviewer 2 Report

The authors still have not fully answer my review comment.

This study design should be improved.
